# REx: Data-Free Residual Quantization Error Expansion

**Edouard Yvinec**[1,2] , **Arnaud Dapogny**[2] , **Matthieu Cord**[1] , **Kevin Bailly**[1,2]
Sorbonne Université[1], CNRS, ISIR, f-75005, 4 Place Jussieu 75005 Paris, France
Datakalab[2], 114 boulevard Malesherbes, 75017 Paris, France
ey@datakalab.com

## Abstract

Deep neural networks (DNNs) are ubiquitous in computer vision and natural language processing, but suffer from high inference cost. This problem can be addressed by quantization, which consists in converting floating point operations into a lower bit-width format. With the growing concerns on privacy rights, we focus our efforts on data-free methods. However, such techniques suffer from their lack of adaptability to the target devices, as a hardware typically only supports specific bit widths. Thus, to adapt to a variety of devices, a quantization method shall be flexible enough to find good accuracy *v.s.* speed trade-offs for every bit width and target device. To achieve this, we propose REx, a quantization method that leverages residual error expansion, along with group sparsity. We show experimentally that REx enables better trade-offs (in terms of accuracy given any target bit-width) on both convnets and transformers for computer vision, as well as NLP models. In particular, when applied to large language models, we show that REx elegantly solves the outlier problem that hinders state-of-the-art quantization methods. In addition, REx is backed off by strong theoretical guarantees on the preservation of the predictive function of the original model. Lastly, we show that REx is agnostic to the quantization operator and can be used in combination with previous quantization work.

## 1 Introduction

Deep neural networks (DNNs) achieve outstanding performance on several challenging computer vision tasks such as image classification [1], object detection [2] and semantic segmentation [3] as well as natural language processing benchmarks such as text classification [4]. However, their accuracy comes at a high computational inference cost which limits their deployment, moreso on edge devices when real-time treatment as well as energy consumption are a concern. This problem can be tackled *via* DNN quantization, *i.e.* by reducing the bit-width representation of the computations from floating point operations (FP) to e.g. int8 (8-bits integer representation), int4, int3 or even lower-bit representation such as ternary (where weights values are either $-1$, $0$ or $+1$) quantization. Because DNN inference principally relies on matrix multiplication, such quantization dramatically diminishes the number of bit-wise operations (as defined by [5]), thus limiting the DNN latency and energy consumption. However, DNN quantization usually comes at the expense of the network accuracy. As a consequence, DNN quantization is an active field of research [6, 7, 8, 9, 10, 11, 12, 13] that aims at limiting this accuracy drop while reducing the number of bit-wise operations.

All the aforementioned methods are data-driven as they either involve training a network from scratch or fine-tune an already trained and quantized one. However, while such approaches usually allow lower quantization errors using low bit-wise representations, due to the growing concerns on privacy rights and data privacy, there is an ever-increasing number of real-case scenarios (e.g. health and

37th Conference on Neural Information Processing Systems (NeurIPS 2023).

military services) where data may not be available for quantization purpose. Furthermore, the bloom of large langage models (LLMs) that are very expensive to train further motivates the use of *post-hoc* data-free quantization methods. Motivated by these observations, several data-free quantization algorithms were published in recent years [14, 15, 16, 17, 18, 19], which focus on the quantization operator, *i.e.* the transformation which maps the floating point weights to their low-bit, fixed point, values. However, these approaches still struggle to offer an interesting alternative to data-driven techniques in terms of accuracy.

Furthermore, when considering a specific target device for deployment, traditional quantization methods, usually focusing on the quantization operator, offer limited options: given a supported bit width (given by the device, as most hardware usually support only a few representation formats [20]) they either achieve satisfactory accuracy or not. To address this concern, we wish to design a flexible quantization method, *i.e.* one that can provide several accuracy *vs.* speed trade-off points for each bit width. Drawing inspiration from wavelets-based methods for image compression [21, 22], we tackle this limitation by considering the successive residual quantization errors between the quantized and original model. Increasing the number of residuals in the expansion (*i.e.* the expansion order) increases the fidelity to the original, non-quantized model at the expense of additional computations. In addition, we propose a group-sparse expansion which allows us to maintain the accuracy using significantly less bit operations. Hence, given a target device, our approach allows finding the best accuracy *vs.* speed trade-off. Our contributions are thus four-fold:

- **REx, a data-free quantization method that is both efficient and flexible.** REx leverages residual quantization, along with group-sparsity, to enable finding suitable trade-offs depending on a target bit-width.

- **Theoretical guarantees** on both the exponential convergence of the quantized model towards the full-precision model and the maximum error with respect to the predictive function. This is of paramount importance in a data-free context, where we cannot easily measure the accuracy degradation.

- **Extensive empirical validation** we show through a thorough empirical validation that, as a standalone method, REx significantly outperforms every state-of-the-art data-free quantization technique, allowing to find better trade-offs on a variety of benchmarks involving ConvNet for classification, object detection or semantic segmentation as well as transformers on GLUE text classification.

- **A ready-to-use solution** that uses a single binary residual to handle outliers within the weight distributions, which is a well-known pitfall when attempting to quantize LLMs.

## 2 Related Work

### 2.1 Quantization

In this section, we review existing methods for DNN quantization, with an emphasis on approaches geared towards run-time acceleration. The vast majority of DNN quantization techniques rely on data usage (Quantization Aware Training). Furthermore, methods such as [7, 8, 9, 10, 11, 23, 24] rely on variants of straight through estimation to alleviate the rounding operation gradients. Among these methods, [25] bears the most resemblance with the proposed REx method. It minimizes the residual error during training, using weight decay over the residue. The similarity with REx comes from the use of a second order expansion of the quantization errors. However, it discards the quantization error after training while we propose to keep the extra operations in order to ensure a high fidelity to the provided pre-trained model.

### 2.2 Data-Free Quantization

Nagel *et al.* [14] discuss the necessity to have data available so as to successfully design a quantization pipeline. They proposed a method that consists in balancing the weight ranges over the different layers of a model, using scale invariance properties that are specific to piece-wise affine (e.g. ReLU) activation functions, and relying on a traditional, naive quantization operator [5]. In the current state of data-free quantization research, we see two major trends: methods that focus on the rounding operator itself [19, 26] and methods that generate synthetic data [27, 12, 13]. With REx, we aim

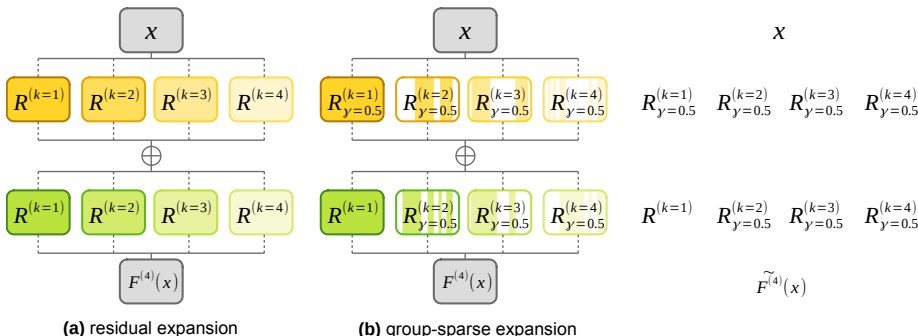

**(a)** residual expansion       **(b)** group-sparse expansion

Figure 1: Illustration of the proposed method for a two-layers neural network. **(a)** residual expansion at order $4$: the intensity of the colormap indicates the magnitude of the residual error. **(b)** group-sparse expansion for orders $k \geq 1$ ($\gamma = 50\%$ sparsity).

at enabling hardware flexibility for these methods by allowing to find better trade-offs in terms of accuracy and compression rate given a fixed bit-width.

### 2.3 Flexibility in Quantization

In practice, the existing data-free quantization methods only offer a single possible quantized model given a supported bit-width. Nevertheless, most hardwares do not support a wide range of bit-width. For instance, Turing [28] and Untether [29] architectures support int4 and int8 quantization while the Nvidia A100 [20] supports int8, int4 and binary (int1) quantization. Conversely, REx circumvents this limitation by offering several trade-offs given a bit-width representation.

## 3 Methodology

Let's consider $F$, a trained network with $L$ layers and trained weights $W_l$. Given a target integer representation in $b$ bits, e.g. int8 or int4, we consider a quantization operator $Q$. Formally, $Q$ maps $[\min\{W_l\}; \max\{W_l\}] \subset \mathbb{R}$ to the quantized interval $[-2^{b-1}; 2^{b-1}-1] \cap \mathbb{Z}$. The most straightforward way to do so is to apply a scaling $s_{W_l}$ and round $\lfloor \cdot \rceil$ the scaled tensor, *i.e.*:

$$Q(W_l) = \left\lfloor \frac{W_l}{s_{W_l}} \right\rceil \tag{1}$$

With $s_{W_l}$ the quantization scale for $W_l$ computed as in [5], without loss of generality. Following the standard formulation [30], a quantization operator $Q$, comes with a de-quantization operator $Q^{-1}$. For the simple quantization operator $Q$ in Equation (1), a natural choice is $Q^{-1}(Q(W_l)) = s_{W_l} \times Q(W_l)$. Note that, despite the notation, $Q^{-1}$ is not a true inverse , as by definition of the quantized space, there is some loss of information. This loss, called the quantization error, is defined as: $W_l - Q^{-1}(Q(W_l))$. In data-free quantization, we want to minimize this error in order to achieve the highest possible fidelity to the original model. In the following section, we describe how we can efficiently reduce the quantization error for a fixed target bit-width $b$.

### 3.1 Residual Expansion

We propose to quantize the residual errors introduced by the quantization process. Although the proposed method can be applied to any tensor, let's consider a weight tensor $W$. In the full-precision space ($\mathbb{R}$), its first approximation is $R^1 = Q^{-1}(Q(W))$. To reduce the quantization error, we define $R^2$ as the quantized residual error

$$R^2 = Q^{-1}\left(Q\left(W - R^1\right)\right) \tag{2}$$

Consequently, during the quantized inference, we compute $R^1 X + R^2 X \approx W X$ which provides a finer approximation than the simple evaluation $R^1 X$. The process can be generalized to any

expansion order $K$, leading to the following:

$$R^K = Q^{-1}\left(Q\left(W - \sum_{k=1}^{K-1} R^k\right)\right) \tag{3}$$

The resulting expanded layer is illustrated in Figure 1 (a) in the case $K = 4$. Intuitively, an expansion $(R^1, ..., R^K)$ provides the approximation $\sum_{k=1}^{K} R^k$ of $W$ and this approximation converges exponentially fast to the original full-precision weights with respect to $K$. As the support of the quantization error space is smaller than one quantization step, the error decreases by a factor larger than $2^b$ with each expansion term (more details in Appendix A). Furthermore, as the quantization error decreases, it is expected that the prediction of the quantized model would achieve a closer match to the original one. This is especially important in the context of data-free quantization as not only do we not have the option to perform fine-tuning to recover accuracy, but also we cannot evaluate the degradation of the model on a calibration/validation set. Nonetheless, we can estimate an upper bound on the maximum error $\epsilon_{\max}$ introduced by quantization on the predictions as

$$\epsilon_{\max} \le U = \prod_{l=1}^{L}\left(\sum_{i=1}^{l}\left(\frac{1}{2^{b-1}-1}\right)^{K-1}\frac{s_{R^i}}{2}+1\right) - 1 \tag{4}$$

where $s_{R^i}$ is the scaling factor from equation 1 applied to each residue. The detailed derivations are provided in Appendix B. This implies that, in practice and regardless on the quantization operator, a network can be quantized with high fidelity with only a few expansion orders to fit a given bit-width. Furthermore, this process can also be applied to the activations.

## 3.2 Input Expansion

Quantizing the weights of a DNN with the aforementioned method already leads to significant memory footprint reduction. However, to significantly decrease the inference runtime, the inputs and activations of each layer also have to be quantized so that each the computations can be processed in the quantized bit-width. For that matter, let $I$ be the input tensor of a layer $l$. We define the expansion of $I$ in quantized residual errors similarly to the weights expansion. Using the generic quantization operator $Q$, we get $I^{(1)} = Q(I)$ and define the $K^{\text{th}}$ order of quantization as

$$I^{(K)} = Q^{-1}\left(Q\left(I - \sum_{k=1}^{K-1} Q^{-1}(I^{(k)})\right)\right) \tag{5}$$

In order to efficiently exploit the resulting expansions, we propose to bound the accumulated order of the weights and inputs. In other words, if we note $k_1$ the expansion order of a residue from the inputs and $k_2$ a residue from the weights, then we only perform the computations for orders such that $k_1 + k_2 < K$ (the rest being negligible in comparison). As a result, the quantized layer $l$ computes:

$$f : I \mapsto \sum_{\substack{k_1, k_2 \in \{1, ..., K\}^2}}^{k_1 + k_2 \le K+1} I^{(k_1)} \otimes R^{(k_2)} \tag{6}$$

where $\otimes$ is the base operation of the layer, e.g. a convolution for a convolutional layer or a matrix multiplication for a fully-connected layer. Similarly to the weights, the error between the full-precision inputs and the proposed expansion of the inputs converges exponentially fast to $0$ with respect to the order $K$ of the expansion. However, with formulations from equations (3) and (5), the overhead computations induced by the expansion is non-negligible. In the following section, we provide a solution to tackle this issue.

## 3.3 Sparse Expansion

The residual expansion as defined in equation 3 is based upon the assumption that the quantization error is equally important for every neuron. Thus, we propose to reduce the overhead cost by only expanding the most important neurons. However, in data-free compression we do not have access to activations or gradients: hence, we measure the relative importance of a neuron in a layer by the norm of its weights [31]. The resulting expanded layer is illustrated in Figure 1 (b). Given a target

budget $\gamma$ (in %) of overhead computations, we only expand the $\frac{\gamma}{K-1}$% most important neurons. The sparse residue is defined as:

$$\left(R_\gamma^{(k)}\right)_i = (R^{(k)})_i \cdot \mathbb{1}_\gamma^{(k)} \tag{7}$$

where $\mathbb{1}_\gamma^{(k)}$ indicates the indices of the most important neurons. Similarly to what precedes, each expansion order is derived sequentially from previous orders and we can bound the quantization error for the sparse expansion (see Appendix A). The method for computing the weights of the expanded model is summarized in Algorithm 1.

---

**Algorithm 1** Expansion Algorithm

---

**Require:** trained DNN $f$ with $L$ layers, hyper-parameters : $K$ and $\gamma$, operator $Q$
  initialize $\gamma^l$ and initialize $f^{(K)}$ as a clone of $f$ with $K$ per-layer kernels
  **for** $l \in \{1, \ldots, L\}$ **do**
    $W \leftarrow$ base kernel of layer $l$ in $f$
    $W_{\mathrm{acc}} \leftarrow 0$ accumulated quantization error
    **for** $k \in \{1, \ldots, K\}$ **do**
      $R_{\gamma^l}^{(k)} \leftarrow Q(W - W_{\mathrm{acc}})\mathbb{1}_\gamma^{(k)}$                     ▶ equation 7
      set $k^{\mathrm{th}}$ kernel of layer $l$ of $f^{(K)}$ with $R_{\gamma^l}^{(k)}$
      $W_{\mathrm{acc}} \leftarrow W_{\mathrm{acc}} + Q^{-1}(R_{\gamma^l}^{(k)})$
    **end for**
  **end for**

---

Also note that in the sparse expansion, we allow higher expansion orders to re-consider neurons that were previously considered unimportant. Consequently, on top of improving the exponential convergence as well as lowering the upper bound on the maximum error with respect to the overhead computations, this method systematically outperforms the standard residual expansion in practice. Proof of this result can be found in Appendix C. The budget $\gamma$ of overhead computations can be set so as not to introduce computational overhead, depending on the bit-width $b$. For example, let's consider a device supporting only 8 and 1 bit (binary) quantization. If we want to achieve the same latency as 8 bit quantization using only 1bit quantization we will have a budget lower than $700\%$ overhead w.r.t. a naive 1 bit quantization. Consequently, for full expansions, we get $\gamma \leq \frac{8}{1} - 1 = 700\%$. This budget is then split across layers using a simple linear repartition. This strategy gives more emphasis to the layers closest to the prediction head which also correspond to the largest layers, and empirically provides the best results [32]. As a result, given a number bit operations (BOPS), the expanded model can better fit the inference device while preserving the full-precision accuracy. Furthermore, all the added computations are performed in parallel which reduces their cost in practice. It allows better trade-offs in terms of accuracy and quantization compression rate, as will be shown in the upcoming experiments.

## 4 Quantization Experiments

In the following sections, we first go through the implementation requirements and efficient strategies to fully leverage the proposed expansions. Second, we perform a comparison of each expansion methods in order to show the flexibility of REx with respect to the bit-width. Third, we compare REx to other quantization schemes under the constraint of equal bit operations. Finally, we validate for each expansion their respective upper bound on the maximum error with respect to the original predictions.

### 4.1 Implementation Details and Benchmarks

We ran our tests on 6 different backbones, including ConvNets and transformers and 5 tasks from both computer vision and natural language processing. We used ImageNet [33], Pascal VOC 2012 [34], CityScapes dataset [35] and GLUE [36] and common sense reasoning benchmarks (details in Appendix D). Unless stated otherwise, we apply symmetric, static, per-channel quantization as defined in [30] and perform batch-normalization folding prior to any processing using the optimal

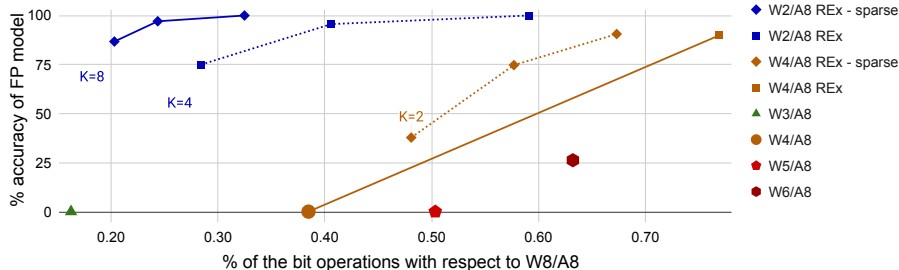

Figure 2: Accuracy *vs.* inference time, for EfficientNet B0. The higher (accuracy) and the further to the left (inference cost) the better. The circles show the baseline results with W3/A8, W4/A8, W5/A8 and W6/A8 quantization. The dashed lines show the trade-offs performance of REx in W4/A8 and ternary quantization (W2/A8). Finally, the plain lines show REx (with sparsity at $10\%$) also in W4/A4 and ternary quantization. The numbers in the symbols stands for the expansion order. REx, and *a fortiori* the sparse version, enables better trade-offs.

method from [37]. In order to leverage the existing efficient implementations of the convolutional layers and fully-connected layers in CUDA, we propose to implement the expanded layer using a single kernel rather than $K$ kernels. This is achieved by concatenating the kernels along the output dimension. Consequently, the challenge of efficiently splitting the computations to fully leverage the target device computational power is left to the inference engine. In practice, this results in both better performance and less work in order to adapt the method to existing engines such as OpenVino [38] and TensorRT [39]. We detail the implementation and overhead of the addition of the residual computations in Appendix E. Furthermore, we evaluate the latency overhead of REx in Appendix F. In the following section, we demonstrate the ability of REx to find good accuracy *vs* speed trade-offs.

### 4.2 Flexible Quantization

Figure 2 shows different trade-offs enabled by REx on different bit-widths for an EfficientNet-B0 on ImageNet. First, the baseline quantization with the baseline quantization operator from [5] (as depicted by the circles of different colors, one for each bit width) offers no trade-off possibility given a specific bit-width and usually performs poorly below int8 quantization (e.g. barely reaching $20.29\%$ top1 accuracy in W6/A8 quantization). REx, however, in the same setup, offers several trade-offs for each specific bit-width (e.g. int4 and ternary on Figure 2) and supporting hardware. Furthermore, the sparse expansion enables finding more potential trade-offs (by varying the budget and expansion order) for every bit-width. Those trade-offs are generally more interesting than comparable ones obtained using the baseline method, which empirically confirms the theoretical results (Appendix C). Furthermore, Figure 2 shows that using higher order, sparse residues allows to find even better trade-offs, as, in this case, e.g. in W2/A8 we reach full-precision accuracy at order 10 with 10% sparse residues. This shows that the process converges fast with respect to the sparsity rates. All in all, these results demonstrate the flexibility of REx to find good accuracy *v.s.* speed trade-offs, given a budget of total bit operations (BOPs) to fit. In the following section, we evaluate the ability of REx to outperform existing quantization methods in terms of equal bops.

### 4.3 Main Results

#### 4.3.1 Experiments on Computer Vision Models

In order to highlight the benefits of residual quantization errors expansions as a stand alone improvement upon existing methods with equal BOPs, we compare REx using the naive quantization operator from [5] on a variety of reference benchmarks. First, in Table 1, we report the performance on three different computer vision networks between state-of-the-art methods in W6/A6 quantization and REx using a sparse expansion at order $K = 2$ using $50\%$ of a 4 bit representation in order to get a similar total number of bit operations (150% of 4 bits $\approx$ 6 bits). For all networks, REx significantly outperforms recent state-of-the-art data-free quantization methods at equal BOPs. Furthermore, we confirm these results on object detection and image segmentation as shown in Figure 3. We can

Table 1: Comparison at equal BOPs with existing methods in W6/A6 and REx with W4/A6 +50% of one 4 bit residue.

| DNN | method | year | bits | Accuracy |
|---|---|---|---|---|
| ResNet 50 (76.15) | DFQ [14] | ICCV'19 | W6/A6 | 71.36 |
| | ZeroQ [17] | CVPR'20 | W6/A6 | 72.93 |
| | DSG [18] | CVPR'21 | W6/A6 | 74.07 |
| | GDFQ [40] | ECCV'20 | W6/A6 | 74.59 |
| | SQuant [19] | ICLR'22 | W6/A6 | 75.95 |
| | SPIQ [26] | WACV'23 | W6/A6 | 75.98 |
| | REx | - | $150\% \times$ W4/A6 | **76.01** |
| MobNet v2 (71.80) | DFQ [14] | ICCV'19 | W6/A6 | 45.84 |
| | SQuant [19] | ICLR'22 | W6/A6 | 61.87 |
| | SPIQ [26] | WACV'23 | W6/A6 | 63.24 |
| | REx | - | $150\% \times$ W4/A6 | **64.20** |
| EffNet B0 (77.10) | DFQ [14] | ICCV'19 | W6/A6 | 43.08 |
| | SPIQ [26] | ICLR'22 | W6/A6 | 54.51 |
| | REx | - | $150\% \times$ W4/A6 | **57.63** |

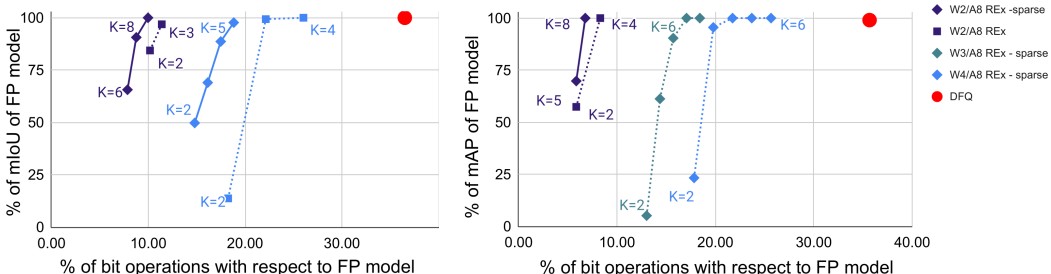

Figure 3: (left) Mean intersection over union (mIoU) of a Deeplab V3+ with MobileNet V2 backbone on CityScapes for semantic segmentation. (right) Mean average precision (mAP) of a SSD with MobileNet V2 backbone on Pascal VOC for object detection. We add the performance of a data-free quantization solution, DFQ [14] for comparison.

observe that REx can maintain the full precision accuracy while dividing by 3.23 the number of bit operations required to run an inference.

### 4.3.2  Experiments on NLP

In Table 2, we perform a similar experiment on NLP using Bert [4]. We can observe the generalization of our results from ConvNets to Transformers, REx can find better accuracy per bits trade-offs as compared to four references including non-uniform quantization [41]. Bert is a pre-trained model with 86 million parameters which is now considered a medium sized model. Both the full-precision and quantized models can fit on a single middle range GPU. However, recent state-of-the-art models, such as OPT [42], are so large that they need multiple gpus just to be loaded on memory. These models ought to be compressed for sustainable usage. In the following section, we generalize the performance of REx to extreme model sizes.

### 4.3.3  Application to Handling Outliers in LLMs

A known pitfall [43] for quantization on LLMs, comes from the presence of extreme outliers among their weight values. These outliers stretch out the weight values range and increase the scaling factor in Equation 1, which, in turn, causes smaller weights to be rounded abruptly to zero. Worse, as suggested in [43], this phenomenon seems to occur more as the model size increases and might appear as a major problem for future work in large DNN quantization. In order to overcome this challenge, we adapt REx to only quantize the outliers in a residue using binary values (W1/A16) while the remaining weights are quantized in int4 (W4/A16). As a result, the overhead from REx is limited to a binary expansion with over 99.8% sparsity. As listed in Table 3, our evaluation on common sense reasoning tasks demonstrates that REx provides a significant improvement over other

Table 2: GLUE task quantized in W4/A8. We consider the BERT transformer architecture [4] and provide the original performance from the article (original) of BERT on GLUE as well as our reproduced results (reproduced). REx is applied to the weights with 3 bits + 33% sparse expansion.

| task | original | reproduced | uniform [5] | log [41] | SQuant [19] | SPIQ [26] | REx |
|------|----------|------------|-------------|----------|-------------|-----------|-----|
| CoLA | 49.23 | 47.90 | 45.60 | 45.67 | 46.88 | 46.23 | **47.02** |
| SST-2 | 91.97 | 92.32 | 91.81 | 91.53 | 91.09 | 91.01 | **91.88** |
| MRPC | 89.47/85.29 | 89.32/85.41 | 88.24/84.49 | 86.54/82.69 | **88.78/85.24** | 88.78/85.06 | 88.71/85.12 |
| STS-B | 83.95/83.70 | 84.01/83.87 | 83.89/83.85 | 84.01/83.81 | 83.80/83.65 | 83.49/83.47 | **83.92/83.85** |
| QQP | 88.40/84.31 | 90.77/84.65 | 89.56/83.65 | 90.30/84.04 | 90.34/84.32 | 90.30/84.21 | **90.50/84.35** |
| MNLI | 80.61/81.08 | 80.54/80.71 | 78.96/79.13 | 78.96/79.71 | 78.35/79.56 | 78.52/79.86 | **79.03/79.96** |
| QNLI | 87.46 | 91.47 | 89.36 | 89.52 | **90.08** | 89.64 | **90.08** |
| RTE | 61.73 | 61.82 | 60.96 | 60.46 | 60.21 | 60.21 | **61.20** |
| WNLI | 45.07 | 43.76 | 39.06 | 42.19 | 42.56 | 42.12 | **42.63** |

Table 3: Evaluation on Common sense reasoning benchmarks for OPT-13B [42] LLM quantized in W4/A16. For each quantization operator DFQ [14], SQuant [19] and PowerQuant [44], we share performance with and without REx (noted with check marks). We also provide the original full-precision (FP) performance.

| | FP | DFQ [14] | | SQuant [19] | | PowerQuant [44] | |
|---|-----|----------|----------|-------------|----------|-----------------|----------|
| **Use REx** | - | ✗ | ✓ | ✗ | ✓ | ✗ | ✓ |
| HellaSwag | 52.43 | 49.25 | **50.14** | 49.23 | **50.21** | **51.29** | 50.98 |
| OpenBookQA | 27.20 | **25.80** | 25.40 | 25.40 | **26.20** | 25.80 | **27.80** |
| ARC-E | 61.91 | 59.93 | **61.91** | 59.97 | **61.95** | **60.82** | 60.52 |
| ARC-C | 32.94 | 30.2 | **32.42** | 30.12 | **32.34** | 31.57 | **32.94** |
| Winogrande | 65.04 | 64.56 | **64.72** | 64.48 | **64.88** | 64.88 | **65.04** |
| PiQA | 76.88 | 75.84 | **76.17** | 75.84 | **76.30** | 75.90 | **76.93** |
| BoolQ | 65.90 | 54.71 | **65.54** | 54.28 | **65.38** | 70.43 | 69.45 |
| Average Score | 54.61 | 51.47 | **53.76** | 51.33 | **53.91** | 54.38 | **54.81** |

Table 4: Upper bound $U$ (see theorem B.1 and B.2) over the maximum error as compared to the corresponding empirical measurement $U_{\text{empirical}}$ of that error for a VGG 16 [45] trained on ImageNet. The closer the upper bound $U$ to the value $U_{\text{empirical}}$ the better.

| weights bit-width | expansion order $K$ | sparsity | $U$ | $U_{\text{empirical}}$ |
|-------------------|---------------------|----------|-----|------------------------|
| 8 | 1 | ✗ | 0.12 | 0.05 |
| 8 | 4 | ✗ | $1.99 \times 10^{-7}$ | $1.78 \times 10^{-7}$ |
| 8 | 2 | 50% | 0.06 | 0.05 |
| 8 | 4 | 50% | $1.17 \times 10^{-7}$ | $0.65 \times 10^{-7}$ |

quantization operators at virtually no cost. Hence, REx appears as a ready-to-use solution to the outlier problem for quantization of LLMs.

## 4.4 Empirical Validation of the Theoretical Bounds

Having shown the interest of REx for quantizing various architectures for computer vision and NLP tasks, we now empirically confirm its mathematical guarantees. In Table 4, we validate the proposed upper bound $U$ in Equation 4 on the maximum error on the predictions on a VGG-16 [45] trained on ImageNet. The tightness of the provided theoretical results can be estimated from the gap between our estimation and the empirical maximum error $U_{\text{empirical}}$ from quantization on the predictions, which is measured as the infinite norm between the full-precision and quantized logits. We observe that a naïve 8-bits quantization (*i.e.* no expansion) leads to an upper bound $U = 0.12$, while we observe $U_{\text{empirical}} = 0.05$. The norms of the logits is equal to $0.3423$. Therefore, the proposed upper bound is relatively tight and significantly lower than the logits magnitude: in such a case, due to overconfidence, the error shouldn't affect the classification. The proposed upper bound is even tighter for larger values of $K$, and becomes lower and lower (for both the theoretical and corresponding empirical maximum errors) when introducing sparsity. This further demonstrates the good properties of the proposed expansion approximation in REx in addition to the relevance of its theoretical guarantees, which are critical in data-free quantization.

Table 5: We report the different trade-offs achieved with REx expanding over different proposed quantization operators in W4/A4 as compared to their performance in W8/A8, on a MobileNet V2.

| method | W4/A4 | W4$_{+\,25\%}$/A4 | W4$_{+\,50\%}$/A4 | W4$_{+\,75\%}$/A4 | W6/A6 | W8/A8 |
|---|---|---|---|---|---|---|
| naive [5] | 0.1 | 53.11 | 64.20 | **71.61** | 51.47 | 70.92 |
| SQuant [19] | 4.23 | 58.64 | 67.43 | **71.74** | 60.19 | 71.68 |
| SPIQ [26] | 5.81 | 59.37 | 68.82 | **71.79** | 63.24 | **71.79** |
| AdaRound [46] | 56.17 | 61.30 | 69.80 | **71.77** | 68.71 | **71.75** |
| BrecQ [47] | 66.57 | 70.94 | 71.28 | **71.76** | 70.45 | **71.76** |

## 4.5 Flexibility with respect to the Quantization Operator

Most recent approaches for data-free quantization focus on designing better quantization operators. Interestingly, as we already hinted on large language models, our approach is agnostic to the choice of the quantization operator and can thus be combined with these approaches without bells and whistles. In Table 5, we report the possible trade-offs achievable with REx combined with recent approaches focusing on the quantization operator on MobileNet V2. The different trade-offs are sorted in ascending order in terms of added overhead operations, e.g. W4$_{+\,25\%}$ leads to less operations than W4$_{+\,50\%}$. First, when used with SQuant [19], REx achieves full-precision accuracy in W4/A4 with only 75% overhead, even outperforming W8/A8 quantization. SPIQ [26], can also be adapted with REx in order to achieve good accuracy using only 4 bits representation as it benefits from finer weight quantization. This explains the slightly higher accuracies than SQuant using 25% and 50% sparsity. Finally, with AdaRound [46] and BrecQ [47], two PTQ techniques, we observe similar results as expected. In particular, BrecQ which already achieves decent accuracy in W4/A4 with a 5.23 points accuracy drop gets closer to the original accuracy (0.86 point accuracy drop) using a quarter of the expansion. Those results demonstrate REx versatility.

## 5 Conclusion

In this work, we proposed a novel data-free quantization method, dubbed REx, that consists in an expansion of residual quantization errors. Furthermore, we proposed a group-sparse version of the residual expansion that allows to find the best accuracy *v.s.* speed trade-offs. We demonstrated the exponential convergence of the quantized weights obtained through the different expansion methods towards the full-precision model. These theoretical guarantees are crucial in the context of data-free quantization where we cannot empirically measure the accuracy degradation in an industrial application context. As such, REx allows to find superior trade-offs for several bit-width representations, which allows better flexibility and adaptability to specific hardwares. In particular, we showed the added value of REx through extensive empirical validation. It appears that REx significantly outperforms recent data-free quantization methods on a wide range of ConvNet architectures applied to image classification, object detection, semantic segmentation as well as transformers architectures on GLUE text classification. Furthermore, we showed that REx allows to efficiently handle outliers within the weight distributions, a well-known pitfall when attempting to quantize LLMs, using a single binary residual to account for outliers. Lastly, the ideas presented in this paper are orthogonal to most recent approaches focusing on improving the quantization operator, and hence can straightforwardly be combined with those approaches.

### 5.1 Limitations:

The residual expansion method introduced in this paper does not adapt to the inter-layer importance and runtime cost discrepancies. An interesting future work would thus consist in applying more expansion orders on the most important layers w.r.t. the model accuracy, as well as using fewer orders for the most computationally expensive layers.

**Acknowledgments:** This work has been supported by the french National Association for Research and Technology (ANRT), the company Datakalab (CIFRE convention C20/1396) and by the French National Agency (ANR) (FacIL, project ANR-17-CE33-0002). This work was granted access to the HPC resources of IDRIS under the allocation 2022-AD011013384 made by GENCI.

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
