# A  Exponential Convergence

The exponential convergence can be proved for the two methods: expansion and sparse expansion. We first prove it for the expansion on sequential models, then generalize the result to more diverse architectures. Before detailing the proof of lemma A.1, we empirically motivate the assumption of symmetry over the weight values distribution. In Figure 4, we plot the distributions of the weights of several layers of a ResNet 50 trained on ImageNet. The assumption is often satisfied in practice. Furthermore, in any instances where it would not be satisfied, it can be enforced using asymmetric quantization.

**Lemma A.1.** *Let $f$ be a layer with weights $W \in \mathbb{R}^n$ with a symmetric distribution. We denote $R^{(k)}$ the $k^{th}$ quantized weight from the corresponding residual error. Then the error between the rescaled $W^{(K)} = Q^{-1}(R^{(K)})$ and original weights $W$ decreases exponentially, i.e.:*

$$\left| w - \sum_{k=1}^{K} w^{(k)} \right| \leq \left( \frac{1}{2^{b-1} - 1} \right)^{K-1} \frac{(s_{R^{(K)}})_i}{2} \tag{8}$$

*where $w$ and $w^{(k)}$ denote the elements of $W$ and $W^{(k)}$ and $(s_{R^{(k)}})_i$ denotes the row-wise rescaling factor at order $k$ corresponding to $w$, as defined in equation 1.*

We work on expanded layers which compute

$$f^{(K)} : x \mapsto \sigma \left( \sum_{k=1}^{K} R^{(k)} Q(x) s_{R^{(k)}} s_x + b \right) \tag{9}$$

*Proof.* Assume $K = 1$, then $W^{(1)}$ is the result of the composition of inverse quantization operator and quantization operator, i.e. $W^{(1)} = s_W \left\lfloor \frac{W}{s_W} \right\rceil$. By definition of the rounding operator we know that $|\lfloor a \rceil - a| \leq 0.5$. Thus we have $|w - w^{(1)}| \leq s_W/2$. Now in the case $k = 2$, we have by definition of the quantization of the residual error and the property of the rounding operator

$$\left| \left\lfloor \frac{w - w^{(1)}}{s_{R^{(2)}}} \right\rceil - \frac{w - w^{(1)}}{s_{R^{(2)}}} \right| \leq \frac{1}{2} \tag{10}$$

where $s_{R^{(2)}}$ is the rescaling factor in the second order residual $R^2$ computed from $w - w^{(1)}$. The quantized weights are thus given by:

$$\left| w - \sum_{i=1}^{2} w^{(i)} \right| \leq \frac{s_{R^{(2)}}}{2} \tag{11}$$

Because the weight distribution is symmetric we know that for any $k$, $s_{R^{(K)}} = \frac{\max\{w - \sum_{k=1}^{K-1} w^{(k)}\}}{2^{b-1} - 1}$ or any other definition of the delta in the full-precision space. Also, by definition we have $\max\{w - \sum_{k=1}^{K-1} w^{(k)}\} \leq s_{R^{(K)}}$. Thus:

$$\left| w - \sum_{k=1}^{K} w^{(k)} \right| \leq \left( \frac{1}{2^{b-1} - 1} \right) \frac{s_{R^{(K)}}}{2} \tag{12}$$

We conclude by using a trivial induction proof. □

As an immediate consequence we have the following corollary which justifies the expansion appellation:

**Corollary A.2.** *Let $f$ be a layer of real-valued weights $W$ with a symmetric distribution and $R^{(k)}$ the $k^{th}$ quantized weight from the corresponding residual error. Then,*

$$\mathbb{E} \left[ \left\| f - \sum_{k=1}^{K} f^{(k)} \right\| \right] \geq \mathbb{E} \left[ \left\| f - \sum_{k=1}^{K+1} f^{(k)} \right\| \right] \tag{13}$$

*and $f = \sum_{k=1}^{\infty} f^{(k)}$.*

The first inequality results from detailing the induction in the previous proof. Instead of an upper bound on the error over all the scalar values we consider each error and show using the same properties that they go down after each step. $f = \sum_{k=1}^{\infty} f^{(k)}$ is a direct consequence of equation 8.

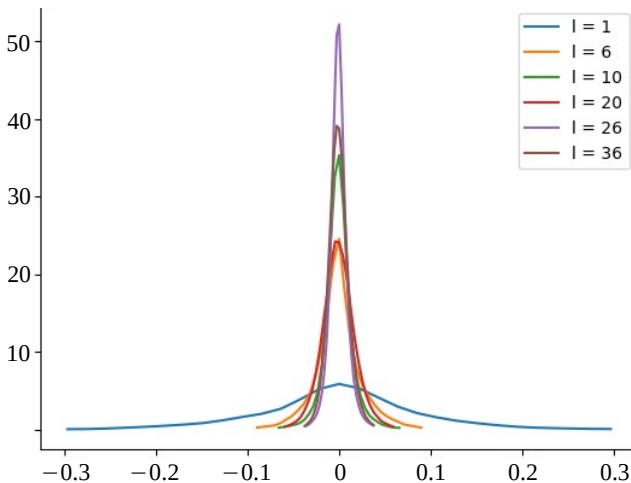

Figure 4: Distribution of the scalar weight values of different layers of a ResNet 50 trained on ImageNet. We observe that every distribution is symmetric around 0.

**Sparse Expansion** Let $N_i^{(k)}$ denotes the $L_1$ norm of an output channel $i$ of the $k$-th order residue $R^{(k)}$. The sparse residue is defined as:

$$\left(R_\gamma^{(k)}\right)_i = (R^{(k)})_i \cdot \mathbb{1}_\gamma^{(k)} \tag{14}$$

where $\cdot$ is the element-wise multiplication, $\mathbb{1}_\gamma^{(k)} = \mathbb{1}_{\{N_i^{(k)} \geq \tau_\gamma^{(k)}\}}$ and $\tau_\gamma^{(k)}$ is a threshold defined as the $\gamma$ percentile of $N^{(k)}$. In other words, we remove a proportion $\gamma$ of channels from residue $R^{(k)}$ that are the least important, as indicated by their norm $N^{(k)}$. Note however that these pruned channels can be encoded in subsequent residuals, *i.e.* $R^{(k')}$, with $k' > k$. The result from Lemma A.1 becomes:

**Lemma A.3.** *Let $f$ be a layer of real-valued weights $W$ with a symmetric distribution. Then we have*

$$\left| w - \left(\sum_{k=1}^{K-1} w^{(k)} + Q^{-1}\left(R_\gamma^{(K)}\right)\right) \right|$$

$$\leq \frac{\left\| N^{(K)} \cdot \mathbb{1}_\gamma^{(K)} \right\|_\infty (s_{R^{(k)}})_i}{\left(2^{b-1} - 1\right)^K 2} \tag{15}$$

*where $\|\|_\infty$ is the infinite norm operator with the convention that $\|0\|_\infty = 1$ and $(s_{R^{(k)}})_i$ denotes the row-wise rescaling factor at order $K$ corresponding to $w$.*

*Proof.* From equation 8, we have:

$$\left| w - \left(\sum_{k=1}^{K-1} w^{(k)} + Q^{-1}\left(R_1^{(K)}\right)\right) \right| \leq \frac{(s_{R^{(K)}})_i}{2} \left(\frac{1}{2^{b-1} - 1}\right)^K \tag{16}$$

which corresponds to the case where $\gamma^l = 1$. If $\gamma^l < 1$, we have two possibilities for $w$. First, the coordinate in $N^{(K)}$ associated to is greater than $\tau_{\gamma^l}^{(K)}$ then we fall in the case where $R_\gamma^{(K)} = R^{(K)}$ and as such we have the result from equation 8 which is stronger than equation 15. Second, the coordinate in $N^{(K)}$ associated to is lower than $\tau_{\gamma^l}^{(K)}$. Then we have that the difference between the baseline weight $w$ and the slim expansion is bounded by the expansion of lower order and the maximum of the norm $N^{(K)}$ which leads to the result in equation 15. $\qquad\square$

**Empirical validation:** In lemma A.1 and A.3 we stated the exponential convergence to 0 of the approximation error on the weight values. In order to empirically confirm this theoretical result, we quantize a ResNet 50 trained on ImageNet in ternary values for different orders $K$. As can be seen in Figure 5, the average error per layer, exponentially converges to 0 which matches our expectations. The figure also confirms the empirical result on the strategies for $\gamma$. The higher errors are located on the last layers, thus these layers require more attention.

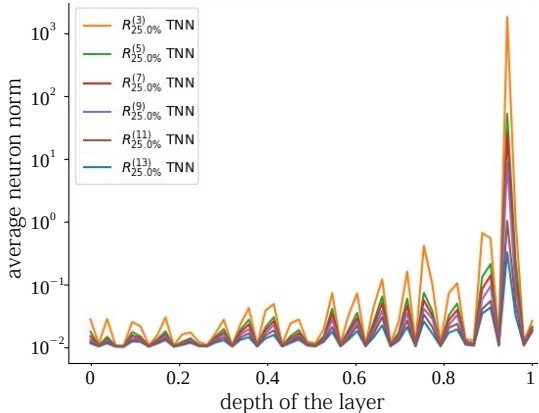

Figure 5: Comparison of the average norm of the quantization error for each layers of a ResNet 50 trained on ImageNet. We observe the exponential convergence stated in lemma A.1 and A.3.

# B   Upper Bound Error

**Theorem B.1.** *Let $F$ be a trained $L$ layers sequential DNN. We note $\sigma_l$ the largest singular value of $W_l - \sum_k R^{(k)}$, i.e. the spectral norm of $W_l - \sum_k R^{(k)}$. Then we have*

$$\max_{\|X\|=1} \|F(X) - F(X)^{(K)}\|_\infty \leq U_{res}$$

$$U_{res} = \prod_{l=1}^{L} \left( \sum_{i=1}^{l} \sigma_i u_i^{(K)} + 1 \right) - 1 \tag{17}$$

*where $u_l^{(K)} = \left( \frac{1}{2^{b-1}-1} \right)^{K-1} \frac{\left( s_{R^{(K)}} \right)_i}{2}$ from equation 8.*

*Proof.* Let's consider $L = 2$, and $F : X \mapsto B\sigma(Ax)$. For any $X$ in the domain of $F$ such that $\|X\| = 1$, we have

$$\|F(X)\|_2 \leq \sigma_B + \sigma_A + \sigma_B \sigma_A \tag{18}$$

where $\sigma_B$ is the largest singular value of $B$ and $\sigma_A$ is the largest singular value of $A$. Following the definition of the 2-norm and $\infty$-norm, we get that

$$\sigma_{A-A^{(K)}} \leq \sigma_A u_A^{(K)} \tag{19}$$

where $\sigma_{A-A^{(K)}}$ is the largest singular value of the residual error of order $K$, $A - A^{(K)}$ and $u_A^{(K)}$ is derived from equation 8. Consequently, we get

$$\|F(X) - F^{(K)}(X)\|_2 \leq \sigma_B u_B^{(K)} + \sigma_A u_A^{(K)} + \sigma_B u_B^{(K)} \sigma_A u_A^{(K)} \tag{20}$$

$\square$

**Sparse Expansion**

**Theorem B.2.** *Let $F$ be a trained $L$ layers sequential DNN. We note $\sigma_l$ the largest singular value of $W_l - \sum_k R^{(k)}$, i.e. the spectral norm of $W_l - \sum_k R^{(k)}$. Then we have*

$$\max_{\|X\|=1} \|F(X) - F(X)^{(K)}\|_\infty \leq U_{sparse}$$

$$U_{sparse} = \prod_{l=1}^{L} \left( \sum_{i=1}^{l} \sigma_i u_i^{(K)} + 1 \right) - 1 \tag{21}$$

*where $u_l^{(K)} = \frac{\left\| N^{(K)} \cdot \mathbb{1}_\gamma^{(K)} \right\|_\infty \left( s_{R^{(k)}} \right)_i}{(2^{b-1}-1)^K 2}$ from equation 15.*

This results is directly derived from Theorem B.1. This result can be extended to more sophisticated architectures. To do so we simply need to address specific attributes such as skip connections, concatenations and other activation functions.

**Skip Connections and Concatenations** In the case of skip connections, the graph is split from a starting layer $l_1$ and split in at least two branches that are added after layer $l_2$ and $l_3$. Assuming we can compute the upper bound for each branch (sub-networks) we simply add these sub-errors. In the case of U-nets, where skip connections contain skip connections, we simply perform this process recursively.

A similar approach can be applied to address concatenations. However in this case we keep the largest value instead of adding them.

**Self-Attention and Cross-Attention blocks** In order to generalize to attention modules, we need to generalize our formula to a product of layers. Let's consider the weight tensors of the keys $W_{\text{keys}}$ and queries $W_{\text{queries}}$. Then the attention scores are computed as follows

$$\text{Att}(X) = (W_{\text{keys}} \times X)^T \times (W_{\text{queries}} \times X) \tag{22}$$

We want to bound the quantization error on the attention mechanism. However, the process involves the magnitude of the inputs $X$ as we highlight

$$\text{Error}_{\text{Att}}(X) = \left\| (W_{\text{keys}} \times X)^T \times (W_{\text{queries}} \times X) - \left( \left( \sum_k R_{\text{keys}}^{(k)} \right) \times X \right)^T \times \left( \left( \sum_k R_{\text{queries}}^{(k)} \right) \times X \right) \right\| \tag{23}$$

If we note $\sigma_k$ and $\sigma_q$ the spectral norms of the residual errors of the keys and queries respectively, then we can simplify the previous formulation

$$\text{Error}_{\text{Att}}(X) = \left\| (\sigma_k \times X)^T (W_{\text{queries}} \times X) + (W_{\text{keys}} \times X)^T (\sigma_q \times X) + (\sigma_k \times X)^T (\sigma_q \times X) \right\| \tag{24}$$

In order to measure this influence on the softmax in the worst case scenario, we can simply compare the $\sigma_k$ and $\sigma_q$ to the smallest singular values of $W_{\text{queries}}$ and $W_{\text{keys}}$. If we note $\alpha_k$ and $\alpha_q$ the largest singular values of $W_{\text{keys}}$ and $W_{\text{queries}}$ respectively, then we get

$$\text{Error}_{\text{Att}}(X)\Big|_{\|X\| \leq 1} \leq \sigma_k \alpha_q + \sigma_q \alpha_k + \sigma_k \sigma_q \tag{25}$$

If we note $\epsilon = \sigma_k \alpha_q + \sigma_q \alpha_k + \sigma_k \sigma_q$ this upper bound, then the error on the softmax scores becomes

$$\text{Error}_{\text{Softmax}}(X)\Big|_{\|X\| \leq 1} \leq 1 - e^{-2\epsilon} \tag{26}$$

**Other Activation Functions** Although ReLU activations are predominant in modern DNNs, there are still many other widely used activation functions such as SiLU, GeLU or even sigmoid. SiLU and GeLU are bounded by the ReLU on the positive side which is where the highest errors occur. Consequently, the upperbound is invariant to GeLU and SiLU activation functions (although under more assumptions on the support, the upper bound could be tightened for ReLU and should be modified for GeLU and SiLU). On the other hand, for sigmoid activations or similar activations (e.g. tanh), the upper bound becomes an upper bound on $X$ in the domain of $F$ instead of $X$ on the unit circle.

## C Sparse Expansion Outperforms Standard Expansion

**Lemma C.1.** *Let $f$ be a layer of real-valued weights $W$ with a symmetric distribution. Then, for $K' < K$ two integers, we have:*

$$Err\left( R^{(1)} + \sum_{k=2}^{K'} R_{\gamma_1}^{(k)} \right) \geq Err\left( R^{(1)} + \sum_{k=2}^{K} R_{\gamma_2}^{(k)} \right) \tag{27}$$

*where Err is the quantization error (i.e. the absolute difference between the quantized and original weights, as in Equation 8) and $K' \times \gamma_1 = K \times \gamma_2 = \beta$.*

*Proof.* Let's assume the layers outputs two channels. Then, we have $\gamma_1 = 1$ and $\gamma_2 = 0.5$. We simply need to prove the result for $k_1 = 2$ and $k_2 = 1$ as the result will extend naturally from this

case. The idea of the proof consists in showing that using lower $\beta$ values enables more possibilities of expansions which may lead to better performance. Let's note $(W)_1$ and $(W)_2$ the weights corresponding to the computation of the first and second output channels respectively. Using $\gamma_1 = 1$, the second order expansion correspond to either quantizing $(W)_1$ or $(W)_2$. Assume $(W)_1$ is chosen for $R_{\gamma_1}^{(2)}$. Then, $R_{\gamma_1}^{(3)}$ will either quantize the error from $(W)_2$ or further quantizes the error from $R_{\gamma_1}^{(2)}$. In the first case we end up with $R^{(1)} + \sum_{i=2}^{k_1} R_{\gamma_1}^{(i)} = R^{(1)} + \sum_{n=2}^{k_2} R_{\gamma_2}^{(i)}$. Otherwise, $\mathrm{Err}\left(R^{(1)} + \sum_{i=2}^{k_1} R_{\gamma_1}^{(i)}\right) > \mathrm{Err}\left(R^{(1)} + \sum_{i=2}^{k_2} R_{\gamma_2}^{(i)}\right)$. $\qquad\square$

## D  Implementation Details and Datasets

We validate the proposed method on three challenging computer vision tasks which are commonly used for comparison of quantization methods. First, we evaluate on ImageNet [33] ($\approx$ 1.2M images train/50k test) classification. Second, we report results on object detection on Pascal VOC 2012 [34] ($\approx$ 17k images in the test set). Third, we benchmark on image segmentation on CityScapes dataset [35] (500 validation images). Our NLP results were obtained on the transfer learning task GLUE [36]. We also evaluate the OPT-13B [42] LLM on the standard common sense reasoning datasets: BoolQ [48], PIQA [49], HellaSwag [50], WinoGrande [51], ARC easy and challenge [52] and OpenBookQA [53]. In our experiments we used MobileNets [54] and ResNets [1] on ImageNet. For Pascal VOC object detection we employed an SSD [2] architecture with MobileNet backbone. On CityScapes we used DeepLab V3+ [55] with MobileNet backbone. We also test our method on VGG 16 [45] and transformers such as BERT model [4] as well as large language models such as OPT-13B [42].

In our experiments, the inputs and activations are quantized using the same method as [14]. We count the bit-wise operations as follows: let $W$ be the real-valued weights of a $d \times d$ convolutional layer on input feature maps of shape $D \times D \times n_i$ and $n_o$ outputs and stride $s$. Then the convolutional product requires $d^2 \frac{D^2}{s^2} n_i n_o$ floating point multiplications. The quantized layer requires two rescaling operations (for the quantization of the inputs and the $Q^{-1}$ operation) and an int-$b$ convolution, i.e. $n_i D^2 + \frac{D^2}{s^2} n_o$ floating point multiplications and $d^2 \frac{D^2}{s^2} n_i n_o$ int-$b$ multiplications. Note that the number of additions remains unchanged. According to [56] the lowest complexity for $b$-digits scalar multiplication is $o(b \log(b))$ bit operations. This is theoretically achieved using Harvey-Hoeven algorithm (also the asymptomatic bound has yet to be proved). We use this value as it is the least favorable setup for the proposed method. As a consequence the number $O_{\mathrm{original}}$ bit operations required for the original layer, $O_{R^{(1)}}$ the number of bit operations for the naively quantized layer and $O_{R^{(k)}}$ for the i$^{\mathrm{th}}$ order residual quantization expansion are

$$\begin{cases} O_{\mathrm{original}} = D^2 \frac{d^2 n_i n_o}{s^2} 32 \log(32) \\ O_{R^{(1)}} = D^2 \left[ (n_i + \frac{n_o}{s^2}) 32 \log(32) + \frac{d^2 n_i n_o}{s^2} b \log(b) \right] \\ O_{R^{(k-1)}} = D^2 \left[ (n_i + \frac{n_o}{s^2}) 32 \log(32) + k \frac{d^2 n_i n_o}{s^2} b \log(b) \right] \end{cases} \qquad (28)$$

Using this result we can estimate the maximum order of expansion before which the number of operations in $f^{(k)}$ exceeds the $O_{\mathrm{baseline}}$. Note that in the case of fully-connected layers, $D = 1$, $s = 1$ and $d = 1$. In the following section, we use the induced metric of accuracy with respect to the total number of bit-wise operations performed by the DNN on a single input. This metric doesn't consider the fact that the added operations can be performed in parallel. For SQuant [19], we use our own implementation which achieve different accuracy results due to different initial accuracies for baseline models. As for ZeroQ [17], we use results provided by SQuant [19]. Similarly to prior work [15, 14, 19], we denote W·/A· the quantization setup (number of bits for weight quantization and number of bit for activation quantization). We used Tensorflow implementations of the baseline models from the official repository when possible or other publicly available resources when necessary. MobileNets and ResNets for ImageNet come from tensorflow models zoo. In object detection, we tested he SSD model with a MobileNet backbone from Manish's git repository. Finally, in image semantic segmentation, the DeepLab V3+ model came from Bonlime's git repository. The networks pre-trained weights provide standard baseline accuracies on each tasks. The computations of the residues as well as the work performed on the weights were done using the Numpy python's library.

Table 6: Overhead induced by the sum reduction in full integer implementation of REx. In this table $R$ means one full residue without sparsity.

| model | W4$_{+25\%}$/A4 | W4$_{+50\%}$/A4 | W4$_{+100\%}$/A4 | W4$_{+1R}$/A4 | W4$_{+2R}$/A4 |
|---|---|---|---|---|---|
| ResNet | 0.035% | 0.070% | 0.138% | 0.415% | 0.830% |
| MobileNet v2 | 0.016% | 0.032% | 0.063% | 0.189% | 0.378% |
| BERT | 0.004% | 0.009% | 0.018% | 0.054% | 0.107% |

Table 7: Overhead induced by the sum reduction in full integer implementation of REx. In this table we study a 1 by 1 convolution on inputs of shape $224 \times 224 \times 320$ and output of 1280 channels as well as a depthwise convolutions on inputs of shape $224 \times 224 \times 96$.

| Filter | expansion order | full runtime (ops) | overhead (ops) | relative cost |
|---|---|---|---|---|
| Conv 1x1 | 2 | $2.0 \times 10^7$ | $6.2 \times 10^4$ | 0.15 |
| Conv 1x1 | 3 | $4.0 \times 10^7$ | $6.1 \times 10^4$ | 0.10 |
| Conv 1x1 | 4 | $6.0 \times 10^7$ | $6.2 \times 10^4$ | 0.08 |
| Depthwise Conv 3x3 | 2 | $2.7 \times 10^5$ | $3.0 \times 10^4$ | 5.49 |
| Depthwise Conv 3x3 | 3 | $5.4 \times 10^5$ | $3.0 \times 10^4$ | 3.38 |
| Depthwise Conv 3x3 | 4 | $8.7 \times 10^5$ | $2.9 \times 10^4$ | 2.43 |

# E    Expansion Reduction in the Accumulator

Let's go though the detailed procedure we applied in order to go from the simulated quantization with floating point scaling factors to integer only inference. We rely on the procedure introduced in [1]. First, let's consider the quantization of a single tensor $A$. Quantization is simulated using $A \approx s_A \times \lfloor A/s(A) \rceil = s_A \times A^Q$ In the present situation, $A^Q$ is quantized and actually fits on the target bit-width while $s_A$ is stored as a floating point value. In order to achieve integer-only inference, we need to convert the multiplication by $s_A$ to an integer multiplication. From [1], we rely on equation (6) and, using similar notations, we get $s_A \times A^Q \approx M_A \times 2^{-n} \times A^Q$. All these operations are integer-only operations. However, in practice, these operations may add errors on top of the quantization scheme itself. To measure this error, we conducted our own experiment and observed that the extra error does not change the quantized output of the layer; this is due to the fact that the term $M_A$ has at least 30 bits of precision while $A^Q$ has 1, 4 or 8 bits of precision (depending on the quantization bit-width). This difference in precision comes from the use of a larger accumulator which is standard in quantized inference.

Now that we detailed how to quantize, we detail how to add two distinct tensors $A$ and $B$ (which will be of special importance to add the residues, as you pointed out). In the simulated quantization, we would get $A + B \approx s_A \times \lfloor A/s(A) \rceil + s_B \times \lfloor B/s(B) \rceil = s_A \times A^Q + s_B \times B^Q$. Now, by applying the same technique as above, we get $s_A \times A^Q + s_B \times B^Q \approx M_A \times 2^{-n} \times A^Q + M_{B/A} \times 2^{-n} \times B^Q$, where $M_{B/A}$ is the integer closest to $M_B/M_A$ encoded with 30 bits of precision. This was discussed in Appendix A.2 in [1]. The authors state that this operation is costly as it requires to perform the integer multiplication prior to the addition. This is a result of the fact that we need to go from the accumulator down to the quantized bit-width and then back up to the accumulator size.

In our pipeline, we limit this cost by using a fused operation in order to introduce low overhead as compared to simply using a larger kernel size. Formally, we used the above mentioned formula directly on the multiplication result. In other words, we get the following formula for $A$ and its quantized residue $R_A$ in the case of quantization using $b$ bits: $s_A \times A^Q + s_{R_A} \times R_A \approx 2^{-n} \times (M_A \times A^Q + M_{R_A} \times 2^{-b} \times R_A)$. Consequently, the overhead from residual summation is limited to a bit-shift on the residue during reduction of the accumulator. In Table 6, we report the relative overhead introduced by this extra bit-shift in the residual summation scheme with respect to the total inference cost. For instance, we list in Table 7 some results with Gap9 hardware: we compare the overhead of computing several convolutional layers with the cost of the reduction of the residuals. On convolutions, as expected the cost are completely negligible. Due to the parallelization abilities of the hardware, as the expansion order increases, the overhead decreases. Furthermore, it should be noted that depthwise convolutional layers are not well supported by most hardware to this day, hence the less impressive results but similar absolute overhead cost.

Table 8: Latency impact of a 50% structured sparsity overhead.

| method | CPU | GPU | accuracy |
|---|---|---|---|
| original model | 7.108 | 0.861 | 76.15 | | | |
| DFQ W6/A6 | 3.039 | 0.606 | 71.36 |
| REx 150% W4/A6 | 3.059 (+0.658%) | 0.564 (−6.930%) | 76.01 |

Table 9: Latency impact of a 1% sparse overhead.

| method | CPU |
|---|---|
| original layer | 0.104799 |
| DFQ W4/A16 | 0.055230 |
| REx W4/A16 + 1% A1/A16 | 0.055376 (+0.2643%) |

# F  Latency Evaluation

**Latency with structured sparse expansions**   50% structured sparsity is leveraged by all hardware devices. Thus, we could conduct our experiments on multiple hardware devices. We considered a CPU (intel xeon) and a GPU (A100) for their difference in bit-width support. The latency is reported as the average over 1000 runs in milliseconds. We decide to compare to DFQ as it is the method that offers the lowest latency due to its use of per-tensor uniform quantization. Our results are listed in Table 8 (for a ResNet-50).

We can observe that although the bops are equivalent REx offers a lower latency than DFQ on GPU by 6.930%. This result can be explained by the fact that in the GPU do provide support in int4 and int8. On the other hand on the CPU, we can clearly see the lack of support for this bit-width which leads to the measurement of the expansion overhead only and an overhead of 0.658%. Still this overhead is fairly limited due to the good parallelization capabilities of the hardware. If measured the throughput instead, the lack of support for the int4 format would hinder the performance of REx on CPU.

Overall, if the hardware supports multiple quantization format, then REx offers the highest accuracy at the lowest inference latency which supports our initial claim that REx offers better trade-offs in terms of accuracy *v.s.* speed.

**Latency with unstructured sparse expansions**   Regarding unstructured sparsity efficiency, we only considered the CPU benchmark as it is the only support for such inference format. For our own sake, we measured the latency of a single fully-connected layer from the MLP block of the OPT-13B model, on which we conducted our initial experiments. Similarly to the previous test, we measure 1000 runs using a naive implementation based on scipy and report the average latency in Table 9.

Our results highlight the marginal overhead of 0.2643% introduced by the sparse binary expansion on fully-connected layers.

All in all, we believe that these results (Table 1 and 2 in this response for latency, as well as e.g. Table 3 in the paper for accuracy on OPT-13B), and the fact that REx significantly improve the accuracy (and outlier handling in LLMs) of existing quantization methods at the price of, in the worst case scenario (when the considered bit-width is not supported), very little latency overhead, and with adequate hardware support, significant speed boost. This further shows the interest of the proposed method.