# OpenReview forum: "REx: Data-Free Residual Quantization Error Expansion"
_NeurIPS.cc/2023/Conference — NeurIPS 2023 poster_

### Official Review · Reviewer_m1LK · 2023-06-30

**Soundness:** 3 good
**Presentation:** 2 fair
**Contribution:** 2 fair
**Rating:** 5
**Confidence:** 3

**Summary:**

The paper proposed REx which allows the flexibility to find the PTQ recipe given a speed/accuracy tradeoff by computing the residual expansion of weights and activations in a data-free manner. The method minimizes error between the FP16 and INT quantizations by computing and adding the quantized residual errors. The authors further propose to reduce the overhead with selective error computations by using the parameter magnitude as a proxy for parameter importance. Additionally, the authors also provide the theoretical error upper bound for a given expansion and show the tightness of the bound.

### Post Rebuttal Update
I have increased my score to 5 -- Borderline accept.

**Strengths:**

The proposed PTQ method is data-free and the idea is quite interesting.

The authors provide theoretical backing for the error upper bound for a given expansion which is greatly appreciated

The paper compares with a fair number of previous approaches


**Weaknesses:**

Figure 2 is extremely difficult to read due to the color scheme (especially for someone who might be colorblind). I would sincerely request to authors to improve the color scheme.

The authors claim that the residual expansions can be made sparse using the norm of the parameters as a selection criterion which might lead to unstructured sparsity. Especially in Table 1, where the sparsity is 50%, this would lead to high inference latencies unless the target device supports unstructured sparsities (which current GPUs do not). Why would this method be preferred over existing methods?

I am not sure I understand the BOPS metric. Hardware targets only support certain bit widths which have a constant cost of operations i.e INT4 cores would take the same amount of time even if you use W3/A3 recipe. I don't think comparing existing methods at equivalent BOPS is fair.

For LLMs, the authors claim that the residue with W1/A16 has virtually no cost. I am not sure I understand I follow this, even with 1bit residue -- Assuming someone has optimized the weight loading with cache-line optimizations that have minimal impact on weight loading times, performing a large number of FP16 operations will add significant computational overhead when the models are compute-bound.

In Line 54, the authors claim a budget of $\gamma$ for the computational overhead but instead use a budget of $\gamma$ for the weight overhead. I would like to point out that a given weight over-head budget does not translate to an equal computational overhead [1], which is usually larger.

In general, there is no discussion on the impact of inference latencies in using this strategy which *significantly* diminishes the impact of this work.

The writing of the paper can be improved. Some of the errors I noticed were: shall -> should in line 8, in -> of in line 83


[1] - https://dl.acm.org/doi/abs/10.1145/3400302.3415679

**Questions:**

I am not convinced about BOPS and how comparing existing methods at equivalent BOPS is fair.

I would like to see some results on latency using this strategy. Latency (and size) of models are the true real-world metrics and I think BOPS is a poor proxy for Latency and limits the applicability of this method.

I would like to understand how the authors propose to deal with unstructured sparsity that the method introduces.

---

> ### Author Rebuttal · Authors · 2023-08-08
>
>
> **figure 2** We updated Figure 2 to make more readable (please see the pdf attached to our general comment)
>
> **bops evaluation and latency** In order to address your major concern regarding performance, we proposed a latency evaluation in the general comment in order to show the good performance of the proposed REx method on both CPU (with limited bit-width support: int8 only) and GPU (supports for multiple bit-widths: int1, int4 and int8). In our original work, we focused on BOPS due to the simpler reproducibility for future research. However, we acknowledge the fact that the proposed REx method does benefit from these new results.
>
> **sparsity: granularity and latency** We think that there were a misunderstanding regarding the sparse expansion and its support. In Table 1 and every other results (aside from LLMs), the introduced sparsity is applied at the neuron level (l154) . Thus, the sparsity is structured and is rather straightforward to leverage on hardware devices. This is confirmed by empirical evidence of latency evaluation (please see Table 1 and 2 in the general comment). On the other hand, regarding the sparse (unstructured) expansion for LLMs, this method does need support for sparse matrix multiplications. While nvidia gpus only support semi-structured pruning we do believe that this format can offer greater benefits as we evaluate it CPU. We show that the sparse binary expansion only adds a 0.26\% overhead in terms of latency which, in our opinion, further supports our initial claim on the negligibility of this operation.
>
> We hope that these clarifications convince you on the interest of the proposed ideas.

---

> > ### Comment · Reviewer_m1LK · 2023-08-14
> >
> > I thank the authors for the clarification. I read through all the other reviews and the authors' replies -- the comparison with Smoothquant is also appreciated. The latency numbers look good and are much more convincing than BOPS.
> >
> > > For LLMs, the authors claim that the residue with W1/A16 has virtually no cost. I am not sure I understand I follow this, even with 1bit residue -- Assuming someone has optimized the weight loading with cache-line optimizations that have minimal impact on weight loading times, performing a large number of FP16 operations will add significant computational overhead when the models are compute-bound.
> >
> > This still looks misleading and should be modified based on the response posted by the authors' describing the specific circumstances in which it is true.
> >
> > > In Line 54, the authors claim a budget of  $\gamma$ for the computational overhead but instead use a budget of $\gamma$ for the weight overhead. I would like to point out that a given weight over-head budget does not translate to an equal computational overhead [1], which is usually larger.
> >
> > Can the authors please clarify this?
> >
> > I would also like the authors to mention the speedup for unstructured sparse expansion is dependent on the target hardware.
> >
> > Based on all the additional evidence, I would like to increase the score to 5.

---

> > > ### Author Response · Authors · 2023-08-14
> > > **Response to Reviewer m1LK**
> > >
> > > We would like to thank the reviewer for their constructive feedback and reactivity. We agree with all the remarks that were made and will make the requested changes, shall the paper be accepted for publication.
> > >
> > > In the specific context of REx, the budget $\gamma$ for weight overhead is very similar to the corresponding computational overhead, due to:
> > >  1. the proposed method increases the layer widths without increasing the output tensors size (due to the reduction from summing the residuals after each layer).
> > >  2. the extra computations have an appropriate structure, e.g. a 50\% overhead from REx on a layer with a power of $2$ neurons, will add a number of computations that is often well supported by the device.
> > >
> > > Empirically, we have measured the induced latency from a small budget $\gamma < 75$% (at least $25$% sparsity) and seen that the parallelization capacities of modern hardware lead to a smaller latency overhead.
> > >
> > > We will also add this element to the implementation section of the revised manuscript.

---

### Official Review · Reviewer_mdRP · 2023-07-04

**Soundness:** 3 good
**Presentation:** 3 good
**Contribution:** 3 good
**Rating:** 5
**Confidence:** 4

**Summary:**

In this paper, the authors propose a fixable quantization method called REx, which utilizes residual error expansion to further improve quantization error. Additionally, they suggest applying group-sparsity to reduce the computational cost associated with residual error expansion.

Existing DFQ methods have a limitation in terms of flexibility when it comes to quantizing based on the representation format supported by hardware, considering the trade-off between accuracy and speed-up. However, the REx method overcomes this limitation and provides a better trade-off. Furthermore, by calculating the theoretical upper bound of quantization error during the process of residual error expansion and demonstrating it through actual quantization error, the REx method shows that it can have less quantization error than or similar to existing methods at fewer BOPs.

The proposed REx method can be combined with recently developed quantization methods, leading to quantized models with improved accuracy.

**Strengths:**

* It demonstrates that the proposed REx method can find a better trade-off point between compression ratio and accuracy compared to existing methods through residual error expansion and group-sparsity expansion.
* Theoretical and experimental evidence is provided for the upper bound of quantization error that the proposed REx method can achieve, showing a theoretical basis for achieving better quantization error at the same BOPs.
* The proposed method can be applied in a composite manner to various quantization methods previously proposed, and experiments on different models demonstrate improvements in accuracy.

**Weaknesses:**

* It appears necessary to compare the results in Table 3 with other papers on LLM quantization, such as OPTQ, LLM.int8, and SmoothQuant.
* Throughout the paper, different bit-widths for weights and activations were used, which may present challenges in utilizing the formats supported by hardware. However, there is a lack of discussion regarding this aspect.
* There is a need for discussion on the criteria for dividing clusters and the benefits of higher sparsity in achieving better accuracy.
* In order to discuss the trade-off between accuracy and speed, there should be a discussion on speed. The paper lacks discussion on this aspect. It would be beneficial to provide unit test results comparing the latency when applying group-sparsity expansion along with INT GEMM + SpMM compared to conventional INT GEMM, as well as the throughput on the GPU.

**Questions:**

1. While Figure 2 represents the inference time in terms of BOPs, is there any analysis regarding the actual latency? I am curious about the analysis of the additional computational cost introduced by residual expansion and how much it can be reduced by group-sparsity expansion.
2. In Figure 2 and Figure 3, when W2A8 is not supported by the hardware, additional packing and unpacking operations are required, which may incur overhead. It would be interesting to see how the graph's trends change when considering this overhead.
3. In Table 5, there is a tendency for higher group sparsity to result in better performance. I am curious about the reasons behind this observation.

**Limitations:**

The paper provided valuable insights by demonstrating the ability of the proposed method to find a better trade-off between compression ratio and accuracy. However, it is challenging to agree with the paper's emphasis on whether it truly offers a superior approach in terms of the accuracy vs. speed trade-off. Since INT GEMM relies on the formats supported by hardware, it is necessary to analyze the speed gain achievable when applying REx + group-sparsity in practice.

---

> ### Author Rebuttal · Authors · 2023-08-08
>
>
> **comparison to other methods on LLMs** In the table below, we provide a comparison between REx (with the PowerQuant method) and other post-training quantization methods designed for LLMs. We report performance in either W4/A16 or W8/A8 based on the available data points excerpted from the papers as well as from our own experiments (Note that, as arguably the biggest bottleneck when quantizing LLMs is memory bandwidth, W4/A16 should be the favored format).
>
> | method | data-usage | average score |
> | :---: | :---: | :---: |
> | DFQ (W4/A16) | data-free | 51.47 |
> | REx + DFQ (W4/A16) | data-free | 53.76 |
> | REx + PowerQuant (W4/A16) | data-free | 54.81 |
> | OPTQ (W4/A16) | data-driven | 54.37 |
> | SmoothQuant (W8/A8) | data-driven | 53.81 |
> | LLM.int8() | data-driven | 54,11 |
>
>
> Our observations are three fold. First, contrary to OPTQ, we do not use group-wise quantization which prevents the quantization of the activations to anything lower than 16 bits. Also, contrary to LLM.int8 we do not need mixed precision between float16 and int8 among the same tensor. And contrary to all of these methods, we do not use any data. Second, OPTQ offers the best performance overall which can be attributed to both the use of group-wise quantization and the weight optimization step. Third, we believe that better performance could be achieved if we were to combine REx and SmoothQuant. Nevertheless, as such, REx+DFQ achieves very close performance to the other methods without suffering from the aforementioned limitations. Furthermore, if we use a recent non-uniform method as the quantization operator rather than DFQ, REx + PowerQuant [4] achieves the highest score.
>
> **actual speed** Thank you for your comment. We used the BOPS metric to provide results that are hardware-agnostic and easier to reproduce and compare with other work. However, we do agree that the method on its own would greatly benefit from actual latency measurements. Please see the general comments (Table 1 and 2) for the latency results.
>
> **hardware bit-width support** As our results suggest in the general comment, when the hardware does not properly support the provided bit-width, packing and un-packing does introduce overhead, which impacts REx performance. In that example, we consider a CPU that only supports int8 and fed it with int4 weight values. The resulting REx model is 0.658\% slower than it DFQ counterpart with 4.65 higher accuracy. All in all, if the hardware offers limited support REx will only improve the accuracy of the final model. However, as our results on the gpu show, if the hardware does support more bit-widths (e.g. int1, int4 and int8), then REx both significantly improves the accuracy.
>
> **higher sparsity in table 5 leading to higher accuracy** There seems to be a misunderstanding here, due to a lack of clarity on our part. The sparsity indicated with respect to the proposed expansion such as W$4_{+ 25\%}$/A4, means that we keep 25\% of the expansion values. In other words, the results provided in Table 5 are indeed intuitive, i.e. the larger the expansion the higher the accuracy. This was stated in l154 where $\gamma$ refers to the kept overhead or in other words the kept values.
>
> ## references
>
> [4] Yvinec, Edouard, et al. "PowerQuant: Automorphism Search for Non-Uniform Quantization." ICLR, 2023.

---

> > ### Comment · Reviewer_mdRP · 2023-08-16
> >
> > Thank you for the detailed answers and results.
> > I have read the authors' rebuttal as well as other reviews. I would like to keep my rating.

---

> > > ### Author Response · Authors · 2023-08-16
> > > **response to reviewer mdRP**
> > >
> > > We would like to thank you for your comment and we are glad to see that we have addressed your concerns. If you were to have any further questions, we would do our best to answer them.

---

### Official Review · Reviewer_d6mY · 2023-07-05

**Soundness:** 3 good
**Presentation:** 3 good
**Contribution:** 2 fair
**Rating:** 6
**Confidence:** 3

**Summary:**

In this work, authors proposed a novel quantization method called REx, that leverages residual error expansion. The idea is to repeatedly apply quantization to the quantized residual error, which provides an increasingly better approximation of the original floating-point tensor as the number of steps grows. To combat the computational overhead from additional quantization steps, authors combine the residual expansion with the group sparsity. Two hyper-parameters of the method (a number of steps K, and the sparsity rate $\gamma$) are used to control the accuracy vs. speed trade-off. The method is agnostic to the quantization operator itself and can be combined with many existing methods.

**Strengths:**

* Experiments on both CNNs (including some of the more difficult-to-quantize models like MobileNets and EfficientNets) and LLMs across various tasks + combination with several quantization operators from the prior work.
* Theoretical derivations on the upper bound of the quantization error + comparison with the empirical values.
* Some attention to efficient hardware implementation (Section 4.1), e.g. instead of using K separate CUDA kernels for the error expansion step, use a single one with concatenated output channels, etc.

**Weaknesses:**

* Two hyper-parameters (K and $\gamma$) that have to be set. Even though $\gamma$ can be set to match a certain BOP target (as it is done in the paper), the user still has to select a value for K.
* I have some concerns related to the data-free nature of the method, please see and elaborate on the below question.
* I would like to see error bars / standard deviations in some of the results, e.g. in Table 2. I wonder if the improvement is statistically significant (it is well known that some of the datasets from GLUE are quite small which might lead to high standard deviations in the results).

**Questions:**

* The method is advertised as data-free, however, I don't see how it is the case in general. As far as I understood, it's data-free if the quantization operator Q itself is data-free (which is the case of DFQ, for instance). However, in most experiments, the underlying Q is not data-free (e.g., AdaRound, BRECQ, or even naive quantization with static range estimation). Could you elaborate?

* Related to the previous comment, in the group sparsity, the importance of a neuron is defined by the norm of its weights (L151: assuming the data-free setting). Assuming we have access to activations and gradients, have you experimented with other importance metrics (e.g., gradient-based, FIT)?

* I Would like to know a bit more details on the outlier quantization in LLMs. How are outliers detected/defined? Will the suggested approach work with outliers that are both positive and negative? In Table 3, what gives the most improvement - the method itself or the special treat of the outliers (would be nice to have an ablation study and/or have some additional data on e.g. quantization error for tensors with outliers with W1 vs. W4).

* (Sorry if I missed it) Do you quantize embeddings, LayerNorm weights/biases, and LLM head, are there any specific assumptions (e.g., first and or last layer in 8-bit)?

**Limitations:**

Authors mentioned one limitation that the method does not adapt to the per-layer importance and runtime cost discrepancies.

---

> ### Author Rebuttal · Authors · 2023-08-08
>
> Thank you for your interest in the method and for pointing out several elements that will improve the clarity of the paper, shall it be accepted for publication.
>
> **set the hyper-parameters** As stated by the reviewer REx uses two hyper-parameters $K$ and $\gamma$. In our main results where we compare REx to other quantization methods, we show that using $K=2$ is almost systemically enough in order to achieve better performance. Aside from LLMs where we use a special setting (unstructured sparsity, removing 99\% of the parameters in the expansion to only account for outliers in said expansion), we use s$50\% tructured sparsity (easier to leverage practically speaking). These set of default hyper-parameters already outperform existing data-free methods. Furthermore, our theory provides that the higher $K$ (and lower $\gamma$ to keep the global budget) the better the performance which indicates that further improvements can be expected when tuning these parameters.
>
> **error bars and significance** We agree on the fact that GLUE benchmark does have variance in the observed accuracies. To answer your concern, we provide here updated versions of our results (we also include other Tables in order to be exhaustive). These changes were added to the revised paper.
>
> | table | Uniform | SQuant | SPIQ | REx |
> | :---: | :---: | :---: | :---: | :---: |
> | updated table 2 (average scores) | 74,16 $\pm$ 0.08 | 74,68 $\pm$ 0.19 | 74,48 $\pm$ 0.35 | 75,00 $\pm$ 0.16 |
>
> Thus, the provided results are significant. On a side note: we did not update the Table 1 and 3 as the quantization process is deterministic and we only have a single pre-trained version of the models.
>
> **is the method data-free:** As stated by the reviewer REx is data-free as long as the quantization operator also is. **Peut etre ajouter qu'on clarifie ça et ajouter 1 ligne ds le papier pour lui donner raison** Aside from Table 5 where we showcase the ability of the proposed method to also work outside of the boundaries of data-free quantization, all of our tests where fully data-free. Thus, we believe that it is appropriate to call our approach data-free as REx in itself do not use data and as such constitute a data-free step in the quantization process.
>
> **other criterion for sparsity** To answer your concern, we tested other critera on ResNet-50 such as _gradients_ and _weights $\times$ gradients_. The results are provided below:
>
> | sparsity method | weights norm | gradients | weights $\times$ gradients |
> | :---: | :---: | :---: | :---: |
> | accuracy (W$4_{+ 25%}$/A4) | 53.11 $\pm$ 0.26 | 52.89 $\pm$ 0.62 | 53.45 $\pm$ 0.37 |
>
> These methods provided small to no improvement, which we attribute to the fact that the sparsity goal here is not challenging enough to justify the use of more complex and costly criterion. We however acknowledge that this is an interesting future research direction.
>
> **how do we identify outliers** We identify outliers among weight values only (contrary to LLM.int8 [2] for example). The reason behind this choice lies in the fact that we do not want to slice tensors at inference and use fine-grained mixed precision, as this generally leads to significant overhead [3]. As we quantize in 4 bits, we define outliers as any value that is more than 6 standard deviations away from the average weight values (Note that this definition is equivalent to the one used in LLM.int8, where the authors define an outlier as any value larger than 6.0 - or $6 \times 1.0$ as features are reduced by the layer norms). Furthermore, in quantization binary values are either $-1$ or $1$, which enables us to quantize both negative and positive outliers.
>
> **LLM specific quantization** To specifically answer your question, we fold the weights and biases from the layer norms, we quantize all the fully-connected layers (including the head) and apply the same quantize to all layers (nothing specific to the first and last layers). We do assume that the softmax and reduction steps are performed in higher precision, as it was done e.g. in I-Bert [1]. **We added those elements in the revised manuscript.**
>
> ## references
>
> [1] Kim, Sehoon, et al. "I-bert: Integer-only bert quantization." International conference on machine learning. PMLR, 2021.
>
> [2] Dettmers, Tim, et al. "Llm. int8 (): 8-bit matrix multiplication for transformers at scale." arXiv preprint arXiv:2208.07339 (2022).
>
> [3] Xiao, Guangxuan, et al. "Smoothquant: Accurate and efficient post-training quantization for large language models." International Conference on Machine Learning. PMLR, 2023.

---

> > ### Comment · Reviewer_d6mY · 2023-08-21
> >
> > Thanks a lot for clarifying my questions and commenting on my concerns. I do appreciate to see that the results are significant and also seeing the actual speed improvement from the general response. I read over all other reviews and the corresponding responses.
> >
> > Overall, considering all strength and weaknesses pointed out by various reviewers, I believe the strength do outweigh some of the weaknesses and therefore still lean towards accept. Therefore I keep my rating of weak accept.

---

### Official Review · Reviewer_E57B · 2023-07-06

**Soundness:** 2 fair
**Presentation:** 3 good
**Contribution:** 3 good
**Rating:** 5
**Confidence:** 4

**Summary:**

This paper presents a low-bitrate post-hoc quantization method with three major techniques: residual expansion, input expansion, and sparse expansion. The techniques are theoretically guaranteed to improve the accuracy of the quantization, and experimental results show that they outperform competing approaches.

**Strengths:**

* Provides theoretical bound for accuracy-bitrate trade-off
* Evaluation that compares the theoretical bound and empirical benchmarks.

**Weaknesses:**

Lack of (machine) runtime evaluation. The project propose to use quantization for better inference runtime, yet none of the benchmark demonstrates the improvement.

**Questions:**

What is the (machine) runtime on CPU and/or GPU?

**Limitations:**

This paper uses public dataset and benchmark, so it would not have potential negative societal impact.

---

> ### Author Rebuttal · Authors · 2023-08-08
>
> We thank the reviewer again. We agree that the proposed study would benefit from an empirical analysis of the actual latency induced by REx. Hopefully the new results provided in the general comment addresses your concern.

---

> > ### Comment · Reviewer_E57B · 2023-08-12
> >
> > It would be helpful if the authors could expand the runtime evaluations to match the benchmark with bops comparisons. The authors indicate that the bops metric is a good proxy for real-world performance, but this connection is not well-supported. Even if the bops metric does provide a good proxy for actual performance, the precise definition of bops is lacking in this work. For future researchers to reference, please provide the formulation of bops.

---

> > > ### Author Response · Authors · 2023-08-12
> > > **Response to reviewer E57B**
> > >
> > > We thank the reviewer for their comment and consideration of the provided runtimes. To address your questions. We use the generic definition of the bitwise operations as in [1,2,3]. This is the standard in quantization and we will include these references to the manuscript for future researcher.
> > >
> > > Regarding the connection between runtime and bops. The goal of the shared results is to highlight that in the context of REx and other standard quantization approaches such as DFQ. The results that were given for equivalent bops (Table 1 in the paper) do translate in very close runtime performance as shown in the common rebuttal. Similarly our results for LLMs (table 3 in the paper) at equivalent bops also translate in almost identical runtime as shown in the second table of the rebuttal. We hope that this message clarifies our previous responses. In short, our rebuttal shows that, in the context of the benchmarks conducted in our study at equivalent bops, the empirical latencies of the proposed REx is indeed almost identical (or even lower) to other quantization schemes while achieving a higher accuracy.
> > >
> > > ## References
> > >
> > > [1] Van Baalen, Mart, et al. "Bayesian bits: Unifying quantization and pruning." Advances in neural information processing systems 33 (2020): 5741-5752.
> > >
> > > [2] Cai, Zhaowei, and Nuno Vasconcelos. "Rethinking differentiable search for mixed-precision neural networks." Proceedings of the IEEE/CVF Conference on Computer Vision and Pattern Recognition. 2020.
> > >
> > > [3] Nikolić, Miloš, et al. "Bitpruning: Learning bitlengths for aggressive and accurate quantization." arXiv preprint arXiv:2002.03090 (2020).

---

> > > > ### Comment · Reviewer_E57B · 2023-08-20
> > > >
> > > > In [1], the authors only count the number of multiplication operations. However, in [3] all bit manipulations are equally attributed.
> > > > > …the BOP count measures the number of multiplication operations multiplied by the bit widths of the operands. To compute the BOP count we use the formula introduced by [3], but ignore the terms corresponding to addition in the accumulator since its bit width is commonly fixed regardless of operand bit width.
> > > >
> > > > Several natural questions arise, such as how to derive the number of BOPs for lookup tables (LUTs) that are commonly used in quantization. The BOPs count for sparse memory loading is also undefined in all these references. Memory operations are much slower than regular bitwise operations on modern architectures, yet there was no discussion on how this is addressed with BOPs. Inconsistent BOPs definitions are a major concern, and the exact formulation was not listed in the supplementary materials. Due to these considerations, I would like to keep my original ratings.

---

> > > > > ### Author Response · Authors · 2023-08-21
> > > > > **response to reviewer E57B**
> > > > >
> > > > > We understand the concern regarding the definition of the BOP count and will add an exact definition to the supplementary material as suggested shall the paper be accepted. Regarding the few points mentioned:
> > > > >  1. on modern hardware, the bit-width of the accumulation is fixed and we do account for it in our measurements due to the reduction operation from REx (this is thoroughly benchmarked in appendix E).
> > > > >  2. the number of BOPs for look-up table is a challenge for non-uniform quantization as we do not of any uniform quantization method that requires LUTs.
> > > > >  3. BOPs with sparsity: we do agree with the fact that this metric is not well suited for sparsity. However, in REx we do not use BOPs to evaluate the unstructured spare expansion which is applied only to LLMs. In that case, we measure the memory foot-print which is the most important challenge to LLM inference and reported latency measurements in our rebuttal which show that empirically, REx achieves state-of-the-art performance by a significant margin in terms of accuracy at the cost of 0.26% of the latency.
> > > > >
> > > > > Overall, the goal of these metrics is to showcase the low cost of REx which offers new trade-offs in terms of accuracy versus speed which is well supported by the combination of our results from the rebuttal with the performance originally listed in the paper.

---

### Author Rebuttal · Authors · 2023-08-08

# General Comment

We thank the reviewers for their interest and critics. The theoretical guarantees and thorough empirical validation offered by REx were highlighted by the reviewers as a strength of this paper. In the initial submission, we measured the accuracy with respect to the number of BOPS (Bitwise Operation Per Second) for a simple and fair comparison. From our understanding, It appears that a major and common concern among some of the reviewers is that we did not provide comparisons in terms of direct on-device latency. We propose a common response to this matter here, and will address the reviewers' other concerns individually.

The choice of the evaluation metric in deep neural network acceleration is not trivial. The BOPS metric bare the advantage of being easily reproducible and comparable as it is (hardware and software agnostic). This is the reason why we opted for this metric. However, we see the point raised by the reviewers and agree that the study would greatly benefit from actual runtime measurements. Consequently, we propose the following extra experiments:
 1. our main result and comparison is reported in Table 1 below, in which we report the latency of the referenced methods as well as the result from REx. Please bear in mind that in Table 1 the sparsity is structured (which we did not clearly mention in the paper).
 2. our second main result is reported in Table 3 with LLMs, in which case the sparsity is unstructured in order to further limit the memory overhead as it is a major concern with LLM quantization. Thus, in the case of LLM only, we will measure the performance obtained using unstructured expansions.

 ## Latency with structured sparse expansions
50\% structured sparsity is leveraged by all hardware devices. Thus, we could conduct our experiments on multiple hardware devices. We considered a CPU (intel xeon) and a GPU (A100) for their difference in bit-width support. The latency is reported as the average over 1000 runs in milliseconds. We decide to compare to DFQ as it is the method that offers the lowest latency due to its use of per-tensor uniform quantization. Our results are listed in the Table below (for a ResNet-50).

| method | CPU | GPU | accuracy |
|:---:|:---:|:---:|:---:|
| original model | 7.108 | 0.861 | 76.15 |
| DFQ W6/A6 | **3.039** | 0.606 | 71.36 |
| REx 150% W4/A6 | 3.059 ($+0.658$) %| **0.564** ($-6.930$) %| 76.01 |

We can observe that although the bops are equivalent REx offers a lower latency than DFQ on GPU by $6.930$%. This result can be explained by the fact that in the GPU do provide support in int4 and int8. On the other hand on the CPU, we can clearly see the lack of support for this bit-width which leads to the measurement of the expansion overhead only and an overhead of $0.658$%. Still this overhead is fairly limited due to the good parallelization capabilities of the hardware. If measured the throughput instead, the lack of support for the int4 format would hinder the performance of REx on CPU.

Overall, if the hardware supports multiple quantization format, then REx offers the highest accuracy at the lowest inference latency which supports our initial claim that REx offers better trade-offs in terms of accuracy *v.s.* speed.

 ## Latency with unstructured sparse expansions
Regarding unstructured sparsity efficiency, we only considered the CPU benchmark as it is the only support for such inference format. For our own sake, we measured the latency of a single fully-connected layer from the MLP block of the OPT-13B model on which we conducted our initial experiments. Similarly to the previous test, we measure 1000 runs using a naive implementation based on scipy and report the average latency in the table down below

| method | CPU |
|:---:|:---:|
| original layer         | 0.104799 |
| DFQ W4/A16             | 0.055230 |
| REx W4/A16 + 1% A1/A16 | 0.055376 $(+0.2643)$% |

Our results highlight the marginal overhead of $0.2643$% introduced by the sparse binary expansion on fully-connected layers.

All in all, we believe that these results (Table 1 and 2 in this response for latency, as well as e.g. Table 3 in the paper for accuracy on OPT-13B), and the fact that REx significantly improve the accuracy (and outlier handling in LLMs) of existing quantization methods at the price of, in the worst case scenario (when the considered bit-width is not supported), very little latency overhead, and with adequate hardware support, significant speed boost. This further shows the interest of the proposed method.

---

### Decision · Program_Chairs · 2023-09-21

**Decision:**

Accept (poster)

**Comment:**

All reviewers voted for "weak/borderline accept". The main reasons were

1. The paper is clearly written.
2. The method is novel, interesting, and with theoretical guarantees.
3. There are extensive experiments, on many models and benchmarks.
4. There are improvements over previous SOTA methods.
5. Some attention is given to efficient hardware implementation.

Though 4+5 could be improved further, I feel these current results are sufficient to warrant acceptence.